

# MaxEnt's parameter configuration and small samples: are we paying attention to recommendations? A systematic review

Narkis S. Morales[1,2,*], Ignacio C. Fernández[2,3,*] and Victoria Baca-González[4]

[1] Department of Biological Sciences, Faculty of Science and Engineering, Macquarie University, Sydney, New South Wales, Australia
[2] Fundación Ecomabi, Santiago, Región Metropolitana, Chile
[3] Landscape Ecology & Sustainability Laboratory, Arizona State University, Tempe, AZ, United States
[4] Facultad de Ciencias Biológicas, Universidad Complutense de Madrid, Madrid, Spain
[*] These authors contributed equally to this work.

Corresponding authors
Narkis S. Morales,
narkis.moralessanmartin@mq.edu.au
Ignacio C. Fernández,
ignacio.fernandez@asu.edu

## ABSTRACT

Environmental niche modeling (ENM) is commonly used to develop probabilistic maps of species distribution. Among available ENM techniques, MaxEnt has become one of the most popular tools for modeling species distribution, with hundreds of peer-reviewed articles published each year. MaxEnt's popularity is mainly due to the use of a graphical interface and automatic parameter configuration capabilities. However, recent studies have shown that using the default automatic configuration may not be always appropriate because it can produce non-optimal models; particularly when dealing with a small number of species presence points. Thus, the recommendation is to evaluate the best potential combination of parameters (feature classes and regularization multiplier) to select the most appropriate model. In this work we reviewed 244 articles published between 2013 and 2015 to assess whether researchers are following recommendations to avoid using the default parameter configuration when dealing with small sample sizes, or if they are using MaxEnt as a "black box tool." Our results show that in only 16% of analyzed articles authors evaluated best feature classes, in 6.9% evaluated best regularization multipliers, and in a meager 3.7% evaluated simultaneously both parameters before producing the definitive distribution model. We analyzed 20 articles to quantify the potential differences in resulting outputs when using software default parameters instead of the alternative best model. Results from our analysis reveal important differences between the use of default parameters and the best model approach, especially in the total area identified as suitable for the assessed species and the specific areas that are identified as suitable by both modelling approaches. These results are worrying, because publications are potentially reporting over-complex or over-simplistic models that can undermine the applicability of their results. Of particular importance are studies used to inform policy making. Therefore, researchers, practitioners, reviewers and editors need to be very judicious when dealing with MaxEnt, particularly when the modelling process is based on small sample sizes.

## INTRODUCTION

Environmental niche modeling (ENM), also referred as to predictive habitat distribution modeling (e.g., *Guisan & Zimmermann, 2000*), or species distribution modeling (e.g., *Elith & Leathwick, 2009*; *Miller, 2010*), is a common technique increasingly used in a variety of disciplines interested in the geographical distribution of species. ENMs have been used, among other disciplines, in landscape ecology (*Amici et al., 2015*), biogeography (*Carvalho & Lama, 2015*), conservation biology (*Bernardes et al., 2013*; *Brambilla et al., 2013*), marine sciences (*Bouchet & Meeuwig, 2015*; *Crafton, 2015*), paleontology (*Stigall & Brame, 2014*), plant ecology (*Gelviz-Gelvez et al., 2015*), public health (*Ceccarelli & Rabinovich, 2015*) and restoration ecology (*Fernández & Morales, 2016*).

The basic principle behind the ENM is the use of environmental information layers and species presence, pseudo-absence or absence points to develop probabilistic maps of distribution suitability (*Elith & Leathwick, 2009*). ENMs are generally used for four main objectives: (1) to estimate the relative suitability of the habitat currently occupied by assessed species, (2) to estimate the relative suitability of habitat in areas where assessed species are currently not known to be present, (3) to estimate potential changes in the suitability of habitat due to environmental change scenarios, and (4) to estimate the species environmental niche (*Warren & Seifert, 2011*).

Among the available tools for ENM, the maximum entropy approach is one of the most widely used for predicting species distributions (*Fitzpatrick, Gotelli & Ellison, 2013*; *Merow, Smith & Silander, 2013*). The maximum entropy approach, part of the family of the machine learning methods, is currently available in the software MaxEnt (*Phillips, Anderson & Schapire, 2006*; https://www.cs.princeton.edu/~schapire/maxent/). MaxEnt can model potential species distributions by using a list of species presence-only locations and a set of environmental variables (*Elith, Kearney & Phillips, 2010*). Since 2004 the use of MaxEnt has grown exponentially (Fig. 1). Nowadays MaxEnt is one of the preferred methods used for predicting potential species distribution among researchers (*Merow, Smith & Silander, 2013*).

The simplicity and straightforward steps required to run MaxEnt seem to have tempted many researchers to use it as a black box despite the increasing evidence that using MaxEnt with default parameter settings (i.e., auto-features) will not necessarily generate the best model (e.g., *Shcheglovitova & Anderson, 2013*; *Syfert, Smith & Coomes, 2013*; *Radosavljevic & Anderson, 2014*). MaxEnt has two main modifiable parameters: (1) feature classes and (2) regularization multiplier. Feature class corresponds to a mathematical transformation of the different covariates used in the model to allow complex relationship to be modeled (*Elith, Kearney & Phillips, 2010*). The regularization multiplier is a parameter that adds new constraints, in other words is a penalty imposed to the model. The main goal is to prevent over-complexity and/or overfitting by controlling the intensity of the chosen feature classes used to build the model (*Elith, Kearney & Phillips, 2010*; *Shcheglovitova & Anderson, 2013*). We recommend look at *Merow, Smith & Silander (2013)* for a detailed explanation of feature classes and regularization multipliers.

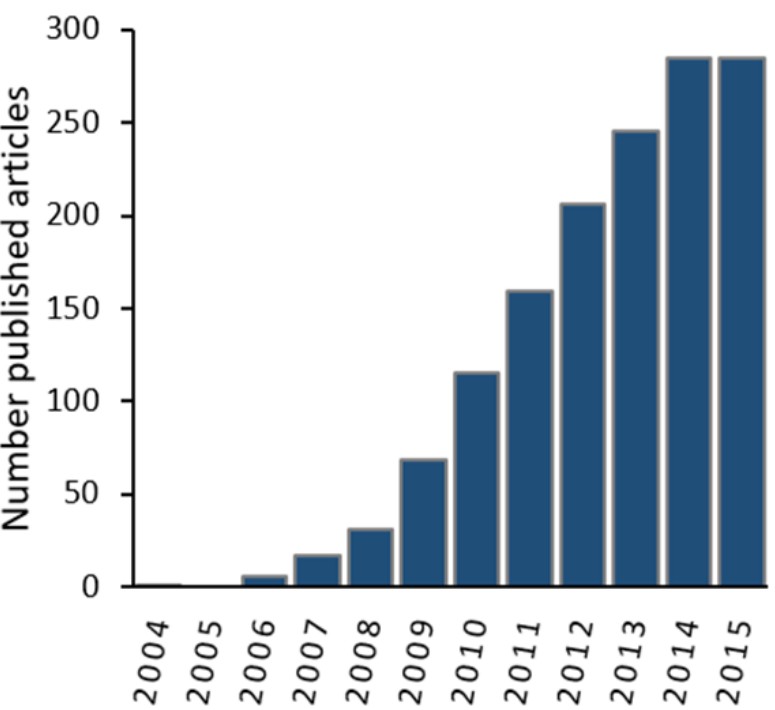

**Figure 1** Number of published articles (2004–2015) containing both "MaxEnt" and "species distribution" within the topic in the Web of Knowledge Databases (see 'Methods' section for databases details).

Some authors have argued that the use of default parameters without providing information on this decision could mean that several of published results could be based on over-complex or over-simplistic models (*Warren & Seifert, 2011*; *Cao et al., 2013*; *Merow, Smith & Silander, 2013*). For example, *Anderson & Gonzalez (2011)* compared different MaxEnt configurations to determine the optimal configuration that minimizes overfitting. Their results showed that in several cases the optimal regularization multiplier was not the default. This is supported by other studies showing that a particular combination of feature classes and regularization multiplier provided better results than the default settings (*Syfert, Smith & Coomes, 2013*), and that the default configuration provided by MaxEnt is not necessarily the most appropriate, especially when dealing with small samples size (*Warren & Seifert, 2011*; *Shcheglovitova & Anderson, 2013*).

Whereas several authors have highlighted the potential problems of models generated by MaxEnt default settings and provided recommendation to deal with this issue (e.g., *Warren & Seifert, 2011*; *Merow, Smith & Silander, 2013*; *Yackulic et al., 2013*; *Halvorsen et al., 2015*), there is no information regarding the echo that these recommendations have had on current MaxEnt use, and neither on how this could be affecting published results. To help answering these questions, in this work we aimed: First, to evaluate if researchers are paying attention to recommendations regarding the importance of evaluating the best potential combination of MaxEnt's parameters for modelling species distribution; and second, to quantify the potential differences in resulting outputs when using MaxEnt default parameters instead of evaluating different sets of parameters combinations to identify an

alternative best model. To achieve our first objective we reviewed and analyzed the published literature from years 2013 to 2015, focusing our analysis in the modelling information provided by articles reporting results based on small numbers of species presence points (i.e., less than 90 presence points). For the second objective, we selected from our review results a sample of 20 case studies, and we performed the modelling process using default setting and a combination of parameters to assess the differences between the default and the alternative best model outputs.

## MATERIALS AND METHODS

### Literature analysis

We used our own literature search protocol using the databases available through the ISI Web of Science (ISI WOS; http://webofknowledge.com/) search engine (Supplemental Information 1) by using the keywords ''MaxEnt'' and ''species distribution'' in the topic. Because many of the recommendations were published between 2011 and 2012, we restricted our search to the 2013–2015 period, assuming that if researchers were alert to recommendations these changes would be noticed on publications of following years. The search was carried out by NS Morales and V Baca-González during the months of March and April, 2016. Whereas we only used English key words for our search, we also included in our analysis the articles published in Spanish and Portuguese but with abstracts written in English. From these results we only selected studies reporting ≤90 presence species points for the modelling process. We chose this threshold value because major changes in MaxEnt auto-features parameters occurs when less than 80 presence records points are used for modelling (*Phillips & Dudík, 2008*; *Merow, Smith & Silander, 2013*), implying that a sample of 90 could easily represent less than 80 presence points for modelling due to the required sample points that needs to be set aside for validation purposes. Because for some authors the ≤90 presence species points threshold may be considered rather large for defining what a small sample size is (e.g., *Phillips & Dudík, 2008*; *Shcheglovitova & Anderson, 2013*), we attempted to overcome this potential issue by ensuring that half of the case studies (i.e., 10) used for performing our modeling analysis had less than 15 presence points.

Our preliminary literature search yielded 816 articles. From these articles, 244 reported a sample size of ≤90 presence points and were therefore used for our literature analyses (Fig. 2, Table 1, see the detailed articles list in Supplemental Information 2). Any doubt or disagreement in the classification of the articles was discussed with IC Fernández; whose opinion was taken as final decision. We reviewed the methodological information provided in the selected articles to determine the types of feature classes and regularization multiplier used for modelling process. We classified features and regularization multiplier used in each paper in three main categories: (1) user-defined parameters, (2) software default parameters, (3) and no information provided. We also evaluated if the articles provided data on the geographical coordinates of presence points used for the modelling process (i.e., lists of georeferenced presence points or species presence maps). In this context, accuracy and precision associated to such geographic information are critical input for performing but are rarely assessed and mentioned in modeling studies (e.g., *Velásquez-Tibatá, Graham*
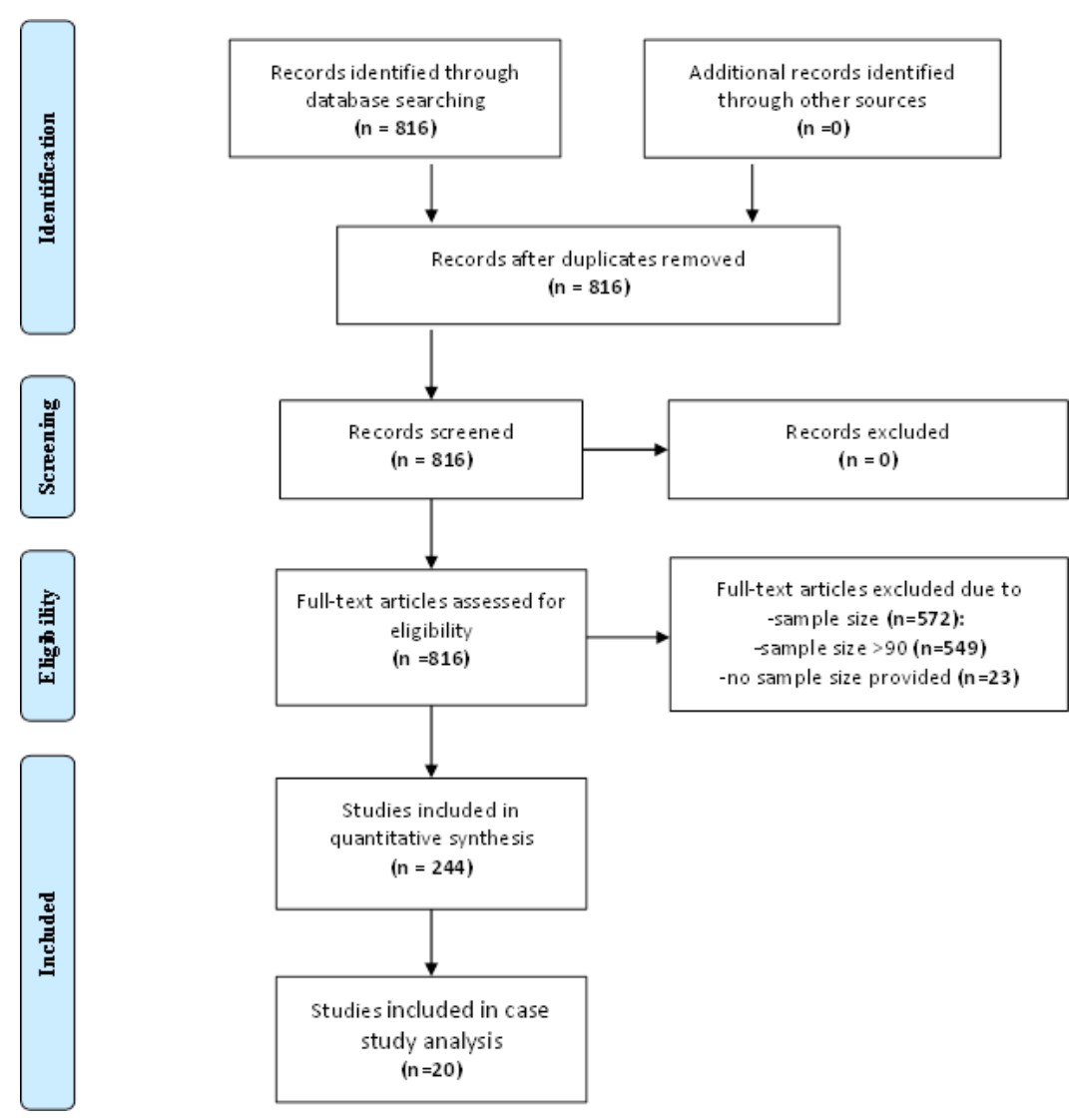

**Figure 2** PRISMA flow diagram of the used search protocol following *Moher et al. (2009)*.

*& Munch, 2016*). We considered only those articles providing information on features, regularization multiplier and geographical coordinates as suitable for modelling analysis.

## Modelling analysis

To quantify the potential differences in resulting outputs when using software default parameters instead of different parameters combinations to identify an alternative best model, we first generated a list consisting on all publications providing the geographical locations of presence points used for modelling and that report having used default parameters (feature classes and regularization multiplier). These selected publications were sorted in two groups, those with less than 15, and those between 16 and 90 sample presence points. From each of these groups we randomly selected 10 articles for our analysis. If the selected articles in any of the two groups were considered too similar in terms of the number of

**Table 1** **Number of articles published during the years 2013–2015 available through the Web of Knowledge Databases.** Articles are presented per year and sample size.

| Year | Total articles | Articles ($n > 90$) | Articles[a] ($n \leq 90$) | Articles (no info) |
|------|----------------|---------------------|----------------------------|--------------------|
| 2013 | 246 | 176 | 65 | 5 |
| 2014 | 285 | 187 | 92 | 6 |
| 2015 | 285 | 186 | 87 | 12 |
| Total | 816 | 549 | 244 | 23 |

**Notes.**
[a]Only articles with sample size $\leq 90$ were used for the analyses.
No info refers to articles that do not provide information about the sample size used for modelling.

samples and area of analysis (extent), we repeated the process until having a heterogeneous sample that increases the strength of our analysis. With this we aimed to include studies from different regions, with varying geographical extents, and differing number of species presence points. For each of these articles we collected the geographical coordinates of species presence points and performed the modelling process using default features, and a set of 72 different parameter combinations, aiming to quantify potential differences on resulting outputs when using default parameters instead of analyzing an alternative best model. For all our modelling we used the WorldClim database (http://www.worldclim.org) as our environmental variables dataset, standardizing all the analysis to a ~1 km$^2$ resolution grid. To select the best model parameters we compared different models with a combination of the "feature class" and "regularization multiplier". MaxEnt provides different types of restrictions ("feature class") in the modelling stage such as lineal (L), quadratic (Q), product (P), threshold (T), and hinge (H). We used all the possible combinations of these features (12 combinations). The used regularization multiplier values were based on *Warren & Seifert (2011)* and *Shcheglovitova & Anderson (2013)*: 1, 2, 5, 10, 15, and 20. Combining features classes and regularization multipliers, we assessed a total of 72 models for each case study, plus the default auto-feature. For each case of study we selected the "best model" by using the AIC$_c$ criterion, as this model selection criterion outperforms other available criterion (e.g., AUC) for comparing different models generated through MaxEnt, particularly for small sample sizes (*Warren & Seifert, 2011*). A detailed description of the methods used for modelling is provided in Supplemental Information 3.

# RESULTS

## Literature analysis

From the 244 articles that reported a sample size ≤90 for the 2013–2015 period, 44.3% (108 articles) did not provide information about the features used for modelling, 39.8% (97 articles) reported to have used default features, and only 16.0% (39 articles) reported to have used user-defined features (Fig. 3; Supplemental Information 2). In terms of the regularization multiplier, 48.8% (119 articles) did not provide any information about the regularization multiplier used for modelling, 43.4% (106 articles) used the default regularization multiplier, and only 6.9% (19 articles) reported having used a user-defined regularization multiplier (Fig. 3; Supplemental Information 2). Considering both default parameters,
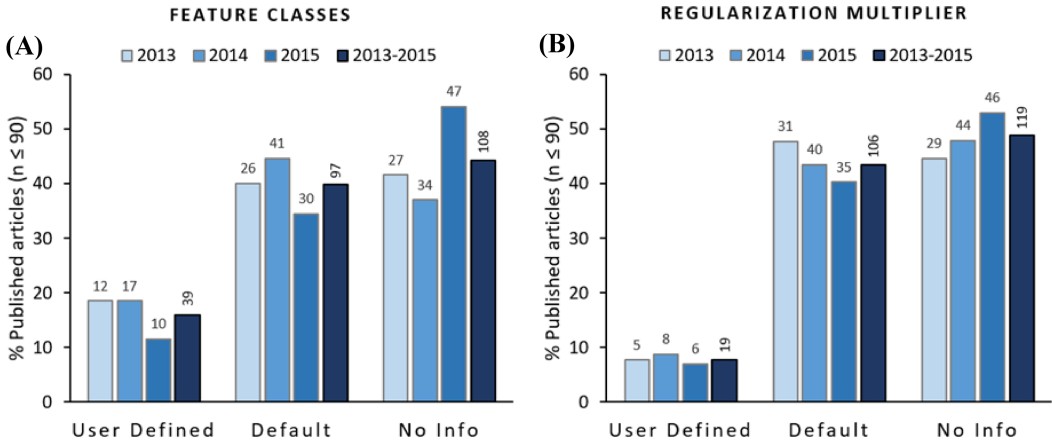

**Figure 3** **Feature classes (A) and regularization multipliers (B) reported to be used for modelling in the analyzed articles.** Columns show the percentage of articles using user-defined, software default, and articles not providing information. Numbers on top of columns represent the number of articles pertaining to each category per year. Columns on the right of each category show the percentage and number of articles for the 2013–2015 period.

merely 3.7% (9 articles) of the reviewed articles reported having used user-defined settings for both parameters (Supplemental Information 2).

Even though 70.5% (172 articles) of publications provide a list or a map with the geographical coordinates of the presence points used for modelling, and 47.1% (115 articles) reported both feature classes and regularization multipliers used for modelling; only 34.3% (84 articles) of the analyzed publications provide all three elements together (Fig. 4).

## Modelling analysis

Results from our modelling analysis reveal huge potential effects of using a default parameter instead of a best model approach for identifying best suitable areas for species distribution (Table 2). Although our results show that the spatial correlation between default and best model outputs is relatively high, and that fuzzy kappa statistics show high similarity between generated maps for all assessed case studies, the total area identified as suitable for the assessed species tend to greatly differ, particularly for species covering large geographical extents (Table 2). Nevertheless, we did not find statistical evidence suggesting that outputs generated by defaults setting tend to predict larger or smaller total suitable areas than the alternative best model ($p = 0.093$, paired $t$-test for log transformed variables). However, it is not only the difference on total suitable area that differs, but also the specific areas that are identified as suitable by both modelling approaches (i.e., shared area). Whereas in average the proportion of shared areas tend to be considerably larger than the not-shared area (mean shared area ratio = 2.483), our data shows that for some cases there could be large discrepancies, with the majority of predicted suitable areas not overlaying between model outputs (i.e., shared ratio < 1) (Table 2).

The sample size (i.e., number of presence points) seems to not affect the degree of differences between the outputs obtained by using the default setting or by evaluating a set of parameters to select an alternative best model. In fact, our analysis show that sample size

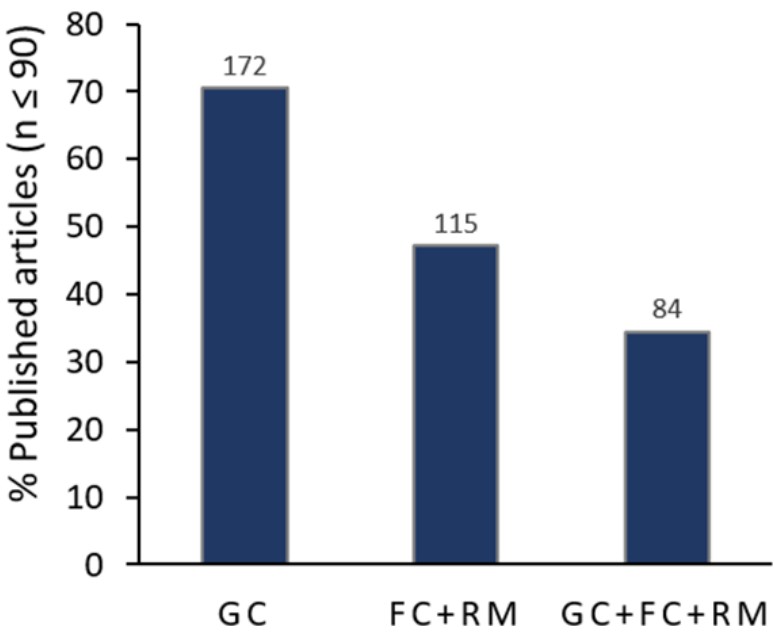

**Figure 4** **Replicability of the modelling process performed in analyzed articles.** Columns show the percentage of articles providing information about GC, geographical coordinates, FC, feature classes, RM, regularization multiplier. Numbers above columns report the number of articles pertaining to each category. Only articles providing information regarding the three inputs (i.e., GC + F + RM column) are considered to provide enough information for replicating the modelling process.

does not affect the spatial correlation ($R^2 = 0.026$, $p = 0.501$), fuzzy kappa ($R^2 = 0.005$, $p = 0.770$), or shared/not shared ratio ($R^2 = 0.004$, $p = 0.786$) between default and best modelling outputs. Also our results do not reveal any trend showing that sample size may favor the selection of some parameter combination over others, beside the fact that for all study cases the best models tend to be associated to small regularization multipliers (Table 2). These results highlight the importance of evaluating what combination of parameters could provide the best modelling results, independently of the sample size used for modelling.

## DISCUSSION

### Are we paying attention to recommendations?

Whereas there is increasing evidence that the use of MaxEnt default parameters do not always generate the best possible model output (e.g., *Syfert, Smith & Coomes, 2013*; *Radosavljevic & Anderson, 2014*), and different authors have highlighted the importance to evaluate the best combination of these parameters before deciding on the best model (see *Anderson & Gonzalez, 2011*; *Warren & Seifert, 2011*), results from our analysis indicate that researchers have been rather indifferent to these recommendations. In fact, our literature analysis shows that the use of MaxEnt default parameters for modelling species distribution with small recorded presence points seems to be the rule rather than the exception. More than 40% of the articles analyzed in our study do not provide information about the parameters configuration used to run the models, which reveals the little attention that researchers and

**Table 2 Estimation of resulting differences when using MaxEnt's default parameters or a best model approach for modelling species distribution.** Spatial correlation values are based in the spatial correlation analysis of MaxEnt's logistic output. Fuzzy kappa was calculated after applying the 10 percentile training presence logistic threshold to generate the species distribution maps. Area values are based on binary maps generated after applying the 10 percentile training presence logistic threshold. Best model parameters represent the combination of feature classes and regularization multipliers of the model identified as of best performance for each study case.

| Sample size | Spatial correlation | Fuzzy Kappa | Area (Km²) | | Area (Km²) | | Shared/not shared ratio | Best model parameters | Source |
|---|---|---|---|---|---|---|---|---|---|
| | | | Default | Best model | Shared | Not shared | | | |
| 7 | 0.856 | 0.864 | 144,129 | 447,092 | 142,612 | 0.466 | 0.466 | T2 | *Carvalho & Lama (2015)* |
| 8 | 0.957 | 0.799 | 76 | 66 | 66 | 6.600 | 6.600 | Q5 | *Fois et al. (2015)* |
| 9 | 0.905 | 0.797 | 15,907 | 9,771 | 9,212 | 1.270 | 1.270 | LQP5 | *Chunco et al. (2013)* |
| 10 | 0.943 | 0.781 | 861 | 1,939 | 843 | 0.758 | 0.758 | Q1 | *Alfaro-Saiz et al. (2015)* |
| 11 | 0.992 | 0.943 | 122,415 | 149,775 | 121,283 | 4.094 | 4.094 | L1 | *Chetan, Praveen & Vasudeva (2014)* |
| 12 | 0.983 | 0.841 | 428,209 | 551,196 | 425,674 | 3.324 | 3.324 | L2 | *Palmas-Pérez et al. (2013)* |
| 12 | 0.836 | 0.906 | 175,166 | 174,543 | 156,798 | 4.342 | 4.342 | TQ5 | *Pedersen et al. (2014)* |
| 13 | 0.960 | 0.843 | 33,421 | 26,169 | 24,317 | 2.219 | 2.219 | TQ2 | *Alamgir, Mukul & Turton (2015)* |
| 13 | 0.995 | 0.965 | 22,013 | 26,445 | 21,820 | 4.528 | 4.528 | LQ1 | *Mweya et al. (2013)* |
| 14 | 0.948 | 0.916 | 363 | 907 | 353 | 0.625 | 0.625 | LQP1 | *Meyer, Pie & Passos (2014)* |
| 15 | 0.967 | 0.900 | 5,004 | 8,845 | 4,991 | 1.291 | 1.291 | QH2 | *Urbani et al. (2015)* |
| 16 | 0.769 | 0.652 | 13,466 | 28,948 | 12,848 | 0.768 | 0.768 | LQPT5 | *De Castro Pena et al. (2014)* |
| 26 | 0.865 | 0.847 | 5,655,316 | 7,383,714 | 5,003,914 | 1.651 | 1.651 | QP1 | *Chłond, Bugaj-Nawrocka & Junkiert (2015)* |
| 26 | 0.945 | 0.705 | 32,020 | 36,420 | 28,695 | 2.597 | 2.597 | L2 | *Simo et al. (2014)* |
| 31 | 0.937 | 0.879 | 243,764 | 248,513 | 196,113 | 1.960 | 1.960 | PT1 | *Orr et al. (2014)* |
| 49 | 0.962 | 0.880 | 135,239 | 103,330 | 100,192 | 2.624 | 2.624 | PT1 | *Hu & Liu (2014)* |
| 54 | 0.945 | 0.858 | 2,491,722 | 1,723,084 | 1,598,103 | 1.569 | 1.569 | LQPT1 | *Conflitti et al. (2015)* |
| 55 | 0.841 | 0.863 | 1,649,518 | 1,570,127 | 1,362,351 | 2.753 | 2.753 | TQ2 | *Vergara & Acosta (2015)* |
| 58 | 0.827 | 0.862 | 5,822,694 | 5,37,0521 | 4,439,531 | 1.918 | 1.918 | T1 | *Aguiar et al. (2015)* |
| 76 | 0.934 | 0.858 | 3,904,018 | 3,700,108 | 3,406,765 | 4.309 | 4.309 | TQ1 | *Yu et al. (2014)* |

reviewers are paying to this specific issue. Our results also reveal that among the articles that do provide information about the features and regularization multiplier used, a large proportion reported to have used the software default configuration. This preference towards using default setting has remained strong despite the variety of articles describing how MaxEnt works and should be used (*Phillips & Dudík, 2008*), the proper configuration process (e.g., *Merow, Smith & Silander, 2013*), the potential implications of not selecting the best parameters combination (e.g., *Anderson & Gonzalez, 2011*; *Warren & Seifert, 2011*; *Syfert, Smith & Coomes, 2013*; *Radosavljevic & Anderson, 2014*) and the increasing publication of approaches to select the best model by using appropriate parameters combinations (see *Anderson & Gonzalez, 2011*; *Syfert, Smith & Coomes, 2013*; *Shcheglovitova & Anderson, 2013*).

We did not observe any trend in the data that would suggest a change from "black box" users towards the use of user-defined parameters. Although our reviewed articles cover a

relatively short period of time (2013–2015), if authors were inclined to adopt best practices for modelling we would have expected to see a trend in the data showing an increasing use of user-defined features over time. However, the only trend in our results is the increasing number of articles not providing information on the features and regularization multiplier used for modelling. We do not have a clear explanation for this trend, but we believe that it is probably due to new researchers using the modelling software without paying proper attention to current MaxEnt literature, particularly to the publications referring to the importance of analyzing parameters combination for selecting the best model (e.g., *Anderson & Gonzalez, 2011*; *Warren & Seifert, 2011*; *Syfert, Smith & Coomes, 2013*; *Radosavljevic & Anderson, 2014*).

The widespread use of default parameters is not the only caveat we found in our literature analysis. We also found a general lack of information that would allow for replicating, assessing or comparing the results from published studies. This information is not only relevant in terms of potential replication of the research, but also necessary for reviewers to evaluate if the outputs from the modelling process are reliable, or are affected among other factors by parameters used, unreliable species presence data sources, or geographically biased presence points records.

Whereas in our literature review we limited the search of articles only to the ISI WOS database, this database includes the large majority of mainstream journals dealing with species distribution modelling (Supplemental Information 2) and is often regarded as including journals with high quality standards. Therefore we consider that our results are a robust representation of the current lack of attention to recent published recommendations on how to better use MaxEnt.

## Implications for research and practice

There are no doubts of the huge potential that MaxEnt has for helping understanding species distribution and for its application as a decision-making tool, which is reflected by the large diversity of disciplines that currently are using it. Nevertheless, as any modelling approach, results obtained through MaxEnt will largely depend on the quality of input data (i.e., reliability of environmental and species presence data) (*Yackulic et al., 2013*) and parameterization used for modelling (*Warren & Seifert, 2011*; *Cao et al., 2013*; *Merow, Smith & Silander, 2013*). Whereas in this work we did not evaluate if the input data used for modelling could be considered reliable or appropriate, it is important to take into account that results can be largely affected by species presence sampling bias (*Kramer-Schadt et al., 2013*; *Syfert, Smith & Coomes, 2013*; *Yackulic et al., 2013*) and by the geographical extent used for modelling (*Merow, Smith & Silander, 2013*).

For the case of parameterization (i.e., combination of features and regularization multiplier), results from our case studies strongly support the claims made by previous studies in relation that using MaxEnt default parameters may not generate the best results (e.g., *Anderson & Gonzalez, 2011*; *Warren & Seifert, 2011*; *Syfert, Smith & Coomes, 2013*; *Radosavljevic & Anderson, 2014*). In fact, in none of the 20 case studies analyzed in our work the model generated by using default parameters were selected as the best model, which is a worrying sign because an important proportion of MaxEnt published literature can be

presenting modelling outputs based on over-simplistic or over-complex models. In other words, reported models can be overestimating the potential distribution of assessed species, or overfitting modelling output to the input data, therefore losing its ability to identify the optimal range of environmental conditions that are suitable for the species (*Warren & Seifert, 2011*; *Merow, Smith & Silander, 2013*).

Nevertheless, perhaps the most relevant implications of an inadequate use of MaxEnt for modelling species distribution are on the decision-making arena. When results from the modelling processes are used directly to assess species conservation or to develop conservation strategies, the areas identified as suitable for a given species could differ greatly depending on the parameters used for modelling (*Anderson & Gonzalez, 2011*). Whereas for our study cases we only used environmental variables gathered from the WordClim database, and therefore our models do not necessary replicate the results published by all the assessed studies, our results do show that independently of the sample size, geographical region and extent of analysis, decisions taken based on models generated by MaxEnt default setting could be strikingly different from those taken based on the best model.

## Conclusions and recommendations

Results from our study may have vast implications, particularly with regard how articles are being reviewed, and the replicability and transferability of the results. We adhere to the calls from other authors to pay better attention to the potential implication of using Maxent's default parameters when modelling species distribution, but we also suggest reviewers to carefully evaluate if the methodological approach used for modelling is reliable and well supported in recent literature. In addition, researchers need to provide as much information as possible to allow proper evaluation and increase the potential replicability and transferability of their results.

Despite the fact that there are several studies that already include several recommendations how to use and set up MaxEnt (e.g., *Elith et al., 2006*; *Warren & Seifert, 2011*; *Merow, Smith & Silander, 2013*; *Yackulic et al., 2013*; *Halvorsen et al., 2015*) we will try to summarize the most important points that researchers need to keep in mind for selecting the best model from the set of potential outputs generated by changing features and regularization parameters.

During the process of building the model, the authors need to determine the best possible model using an objective methodology. One approach is using a jackknife procedure similar to the one described by *Shcheglovitova & Anderson (2013)*. The process consists in comparing different models with a combination of the parameters, "feature class" and "regularization multiplier" (see *Shcheglovitova & Anderson (2013)*, *Warren & Seifert (2011)* and Supplemental Information 3 for examples). The comparison of models can be done using the corrected Akaike information criterion (AICc) available in the software ENMTOOLS version 1.4.4 (*Warren & Seifert, 2011*). The best model will correspond to the combination of "feature class" and "regularization multiplier" with the smallest AICc value. Although this is the methodology that we used in this work there are other methods that can be used. Another option is using a correlation analysis of the model-predicted probabilities of occurrence and presences and absences proposed by *Syfert, Smith & Coomes (2013)*

or comparing the different map outputs using the fuzzy kappa statics based on *Mestre et al. (2015)*.

Once the best model is selected, replication of the best model (e.g., several runs; $n = 30$) is needed to determine that the results are consistent. Also, it is highly recommended to validate the model output using *in situ* surveys especially in cases that small numbers of occurrences were used to generate the model. Although, we understand that this could be a major task when modelling large extensions of habitat or rare species distributions; these limitations must be included in the discussion and used with caution, especially for management purposes.

These simple recommendations can help to improve the applicability of resulting models, which in turn will help practitioners and decision-makers to use them more effectively as practical tools for the development of management and conservation activities. While the use of MaxEnt's default parameter can be very useful for having a quick picture of the potential distribution of a given species, taking the necessary time to evaluate which parameters combination results in the best model could largely increase the accuracy and reliability of modelling results.

### Funding
The authors received no funding for this work.

### Competing Interests
The authors declare there are no competing interests.

### Author Contributions
- Narkis S. Morales and Ignacio C. Fernández conceived and designed the experiments, performed the experiments, analyzed the data, wrote the paper, prepared figures and/or tables, reviewed drafts of the paper.
- Victoria Baca-González performed the experiments, analyzed the data, reviewed drafts of the paper.

### Data Availability
The raw data is included in the manuscript and supplied as Supplemental Files.

### Supplemental Information
Supplemental information for this article can be found online at http://dx.doi.org/10.7717/peerj.3093#supplemental-information.

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
