# Peer review of "MaxEnt’s parameter configuration and small samples: are we paying attention to recommendations? A systematic review"

_PeerJ, doi:10.7717/peerj.3093_

## Round 0.1 · original submission · Major Revisions

All three reviewers were positive, but they had also critical comments which need to be addressed carefully. In particular, I would emphasize that your paper would have more impact if you can come up with some recommendations.

Reviewer 1 ·

Basic reporting

The abstract should highlight that you reanalyzed a number of studies and found different results - that will tempt readers to keep reading and find out how they differ.

Summarize which are the most critical parameters to explore, how best to modify parameters in your opinion, briefly and provide a guide on where to find more details in the conclusions. As it stands, the study accurately describes poor modeling practice well but doesn't use that as leverage to steer those modelers in the right direction. The impact of the article could be much stronger by including some brief recommendations.

Experimental design

124 - It's great to leave the gritty details of methods to the appendix, but it would be better to provide a 1 paragraph summary of how you modified parameters. Keep in mind that people maybe not know how to best explore parameters so this isn't *just* methods, but also your recommendation for how others should do it.

Validity of the findings

Well supported.

Additional comments

Morales et al provide a clear and concise description of one of the major problems in biogeography. The results are well founded and they do a nice job of providing an objective description of other researchers' poor modeling practices without being tempted to offer much judgement, which I think is a very professional way to get readers attention (in contrast to some previous Maxent criticisms). My primary criticism is that they do not leverage their findings to advance the field by providing some guidance on how researchers should proceed. As this is a review, it's not critical that they come up with their own suggestions, but rather provide a short field guide to the existing literature. It may simply be that the researchers they criticize are overwhelmed with the enormous maxent literature. Hence I *strongly* recommend adding a section to the discussion to summarize which are the most critical parameters to explore, how best to modify parameters in your opinion, and briefly guide on where to find more details.

93, 102: unnecessary detail

- Its worth mentioning the study on sampling bias by Yackulic in 2013 and how that also suggests that results are invalid if its ignored.

Reviewer 2 ·

Basic reporting

In this manuscript, an interesting review is presented by Morales et al. to test if MaxEnt users are paying attention to the software’s recommendations. It is remarkable the work performed and the amount of data collected. In my opinion this is an interesting manuscript, well written, on an important topic.

Nonetheless, is my opinion that the manuscript in its current form falls short in several key-points that must be addressed carefully before this manuscript can be accepted for publication in any scientific journal. The most important key-points that, in my point of view, must be addressed are signalled by *

Experimental design

Introduction:
L46 – species distribution models are used in ecology not only to predict but as well to explain, therefore not only “disciplines that use spatial-explicit ecological data” applies this type of models. In fact, MaxEnt provides direct spatial explicit results, but the majority of platforms and packages don't. I suggest the authors modifying this along the whole manuscript.

L53 replace “…pseudo-absence and absence…” by “… pseudo-absence or absence…”

L53 (but though all the introduction as well) – the models calibrated using species distribution models are not always probabilistic. If you calibrate your model with presences and absences without knowing the real ratio between presences-absences in the whole range of the species (i.e. often), the results must be presented as presence and absence as well. A possible solution often used by modellers, it to present results below the threshold as absences and then present a probabilistic map above the threshold value. So, the authors must consider changing this along the text (basically not focus on the probabilistic outputs).


L62* The reference Merow et al. 2013 does not support the written sentence. To support the reference the authors must find a study (or perform an exploratory analyses themselves) with a meta-analyses about the use of different modelling techniques in species distribution modelling along time.

L69-70 “… over-complex or over-simplistic models…” – this issue must be deepest explored in the introduction section, i.e. the problems and impacts regarding the use of wrong or not optimized parameters in modelling.

L82* – why 90 is a small number of presences? The authors must support this option very well. In my point of view, 90 presences can represent a large number of occurrences if the study area is small and if the occurrence captures the whole environmental range for the species in that area. Some publishes studies consider for modelling purposes a small number of presences being 10, but I don't know any study that states 90 as a small number of presences. In my point of view this choice, even with the justification given in L96-99, is not clearly supported and can be a huge fragility of this manuscript as the small/big number of occurrences is very scale dependent. The authors must explore this scale-dependence issue along the whole ms.

I have missed more clear objectives definition with goals and hypothesis (to test them) in the end of the introduction.

Methods:
Literature analysis

L91 – In this type of literature analyses the best protocol in my point of view is the PICO (population-intervention-comparison-outcome), it has already been tested in several meta-analyses and review papers. I suggest the authors to test this protocol as well.

L92 – The database must be stated as: ISI Web of Science (ISI WOS; http://webofknowledge.com/)

*In this type of reviews there are 3 “mandatory” platforms, ISI WOS, Scopus and Science Direct – the authors must perform the same search in those 2 platforms missing and analyse the outcomes.

*After doing the search, and to evaluate the reliability of the search, the first 50 records retrieved by Google Scholar (using the same keywords) must be checked and analysed as well.

*2013-2015 period – even if the recommendations about MaxEnt were only published between 2011 and 2012, this period must be considered as well. The authors must perform the search between the date the software was released and 2016, and see if the recommendations changed anything about the published modelling options.

*L96-98 – if the authors chose the 90 as the threshold between small and large presences samples they must find some references to support their choice and explore the scale dependency along the manuscript. I’m not confortable with this threshold value as I don’t know any scientific paper to support it, and there are some papers stating that MaxEnt is really suitable to apply to a small number of presences, and their small number is not 90…

The authors must check these papers, they are very important to include in the review and support (or not!) some choices:

Elith et al. 2010 - http://onlinelibrary.wiley.com/doi/10.1111/j.1472-4642.2010.00725.x/full, and

Elith et al. 2005 Novel methods improve prediction of species’ distributions from occurrence data


L102-103 “Any doubt …. decision” I suggest the authors eliminating this sentence.

*The manuscript misses a sub-section about the inclusion/exclusion criteria – “with reasons” like presented in Figure 2 cannot be accepted. All “reasons” must be stated.

*The consistency of results must be assessed through kappa statistics on 10 % of randomly chosen records (see Higgins and Green, 2011 for details)

Higgins, J. P. and Green, S. 2011. Cochrane Handbook for Systematic Reviews of Interventions Version 5.1.0. The Cochrane Collaboration, 2011. Available from http://handbook.cochrane.org.


Comparison of default and best model

L117 – “selected 20 articles” – how that was performed, randomly?

L119 – some information about the selected studies must be included as a table in the main text: the regions, the geographical “extentions” (replace extentions by extents) , and number of presence points.

Validity of the findings

Results
L142 – this subsection merges results with discussion – the authors must move the discussion around the results to the discussion section.

*Results must be updated after considering the suggestions for the manuscript improvement.

Discussion:
*Subsections must be added to the discussion, copping with the manuscript objections.
The discussion is to vague and the novelty for science as well as the main findings must be highlighted in the text.

Additional comments

No comments

·

Basic reporting

Introduction: Although feature classes and regularization multiplier are defined in previous studies (e.g. Syfert et al. 2013), a section introducing these two parameters and their impacts within the modeling process would make the paper clearer.
Results:
Line 152: “geographical coordinates”. I am not sure to understand what the authors mean here: is it related to the precision / accuracy of locations or related to spatial auctocorrelation issues?
Figures: I find the figure 2 not informative as a main figure. I would move it to the supplementary material.

Experimental design

I found the analysis performed in table 2 particularly interesting and well explained. I think it is the key result of this study. However, I am not sure to understand why the authors used the 10 percentile training presence logistic threshold (only) instead of complementing with other reclassification / optimization methods such as MaxAUC or MaxTSS (?)

Validity of the findings

My only concern is related to the use of one reclassification threshold only in table 2 (10 percentile training presence logistic threshold) instead of providing several widely-used techniques such as MaxAUC or MaxTSS.

Additional comments

Review of manuscript # 13850 “MaxEnt’s parameter configuration and small samples: Are we paying attention to recommendations? A systematic review” by NS Morales et al. for PeerJ

In this manuscript, Morales et al. reviewed 244 articles from 142 journals to assess whether modelers are really following recommendations to avoid using the default parameter configuration when using MaxEnt with small sample size to calibrate environmental niche models (ENMs). Based on the study by Syfert et al. (2013), they focused on two parameters, which were feature classes (environmental response curves) and regularization multiplier (related to constraints and variable selection). The authors found that the two parameters were properly evaluated separately or simultaneously in only a small proportion of the studies considered in this meta-analysis.

General comments

This manuscript is concise, in general clear and well written and overall based on a nice idea: as stated by the authors, ENM projections are ultimately used as decision tools to inform policy making and there is thus a need provide robust and safe projections. Since ENMs and in particular MaxEnt are now widely used in ecology, this study can potentially reach a broad audience of ecological modelers and biogeographers.
I have however a set of minor comments that I hope will help to further improve this nice manuscript.

Specific comments
Introduction: Although feature classes and regularization multiplier are defined in previous studies (e.g. Syfert et al. 2013), a section introducing these two parameters and their impacts within the modeling process would make the paper clearer.
Results:
Line 152: “geographical coordinates”. I am not sure to understand what the authors mean here: is it related to the precision / accuracy of locations or related to spatial auctocorrelation issues?
Figures: I find the figure 2 not informative as a main figure. I would move it to the supplementary material.

I hope these comments will help in improving this nice manuscript.

---

## Round 0.2 · accepted · Accept

One referee had only a minor suggestion regarding one sentence of the paper that I think is worth including while in production. There are papers looking at the consequences of location error (eg Velásquez-Tibatá, J., Graham, C.H. & Munch, S.B. (2016) Using measurement error models to account for georeferencing error in species distribution models. Ecography, 39, 305-316) that you could refer to in this context.

·

Basic reporting

As in the previous version, the MS is clear and i general well written.

Experimental design

The methods have gained in clarity since the first version.

Validity of the findings

This paper will be useful for a broad audience of species distribution modelers in ecology.

Additional comments

My comments have been very carefully addressed and the MS gained in clarity in the methods.
I have only one last comment regarding the geographic coordinate issue:
Line 150 of the new version:
"We also evaluated if the articles provided data on the geographical coordinates of presence points used for the modelling process (i.e. lists of georeferenced presence points or species presence maps)"
I would change to:
"We also evaluated if the articles provided data on the geographical coordinates of presence points used for the modelling process (i.e. lists of georeferenced presence points or species presence maps). In this context, accuracy and precision associated to such geographic information are critical input for performing but are rarely assessed and mentioned in modeling studies."

Well done!